# Use of an Innovative Silage of Agro-Industrial Waste By-Products in Pig Nutrition: A Pilot Study of Its Effects on the Pig Gastrointestinal Microbiota

**DOI:** 10.3390/microorganisms11071723

**Published:** 2023-06-30

**Authors:** Ioannis Skoufos, Aikaterini Nelli, Brigkita Venardou, Ilias Lagkouvardos, Ilias Giannenas, Georgios Magklaras, Christos Zacharis, Lizhi Jin, Jin Wang, Evangelia Gouva, Stylianos Skoufos, Eleftherios Bonos, Athina Tzora

**Affiliations:** 1Laboratory of Animal Science, Nutrition and Biotechnology, Department of Agriculture, School of Agriculture, University of Ioannina, Kostakioi Artas, 47100 Arta, Greece; gmag@uoi.gr (G.M.); x.zaxaris@uoi.gr (C.Z.); ebonos@uoi.gr (E.B.); 2Laboratory of Animal Health, Hygiene and Food Quality, Department of Agriculture, School of Agriculture, University of Ioannina, Kostakioi Artas, 47100 Arta, Greece; k.nelli@uoi.gr (A.N.); venardou@gmail.com (B.V.); ilias.lagkouvardos@tum.de (I.L.); egouva@uoi.gr (E.G.); steliosskoufos@gmail.com (S.S.); tzora@uoi.gr (A.T.); 3Laboratory of Nutrition, School of Veterinary Medicine, Faculty of Health Sciences, Aristotle University of Thessaloniki, 54124 Thessaloniki, Greece; igiannenas@vet.auth.gr; 4Meritech (Asia Pacific) Biotech Pte Ltd., Singapore 079903, Singapore; jin@meritech.com.cn; 5National Center for International Research on Animal Gut Nutrition, Jiangsu Key Laboratory of Gastrointestinal Nutrition and Animal Health, Laboratory of Gastrointestinal Microbiology, College of Animal Science and Technology, Nanjing Agricultural University, Nanjing 210095, China; jwang8@njau.edu.cn

**Keywords:** olive mill by-products, winery by-products, dairy by-products, 16S rRNA sequencing, gut microbiota, *Clostridium*, *Streptococcus*

## Abstract

The aim of this study was to evaluate whether dietary supplementation with an innovative silage (IS) created using 60% olive mill waste, 20% grape pomace, and 20% deproteinised feta cheese waste solids can modulate the composition of the intestinal microbiota in weaned (Exp. 1) and finishing (Exp. 2) pigs. In Exp. 1 (40 day supplementation), forty-five crossbred weaned pigs were randomly assigned to the 0% (Control), 5%, or 10% IS groups (15 replicates/experimental diet). In Exp. 2 (60 day supplementation), eighteen finishing pigs from Exp. 1 were fed the control diet for 8 weeks before being re-assigned to their original experimental groups and fed with the 0% (Control), 5%, or 10% IS diets (six replicates/experimental diet). Performance parameters were recorded. Ileal and caecal digesta and mucosa were collected at the end of each experiment for microbiota analysis using 16S rRNA gene sequencing (five pigs/experimental diet for Exp. 1 and six pigs/experimental diet for Exp. 2). No significant effects on pig growth parameters were observed in both experiments. In Exp. 1, 5% IS supplementation increased the relative abundance of the *Prevotellaceae* family, *Coprococcus* genus, and *Alloprevotella rava* (OTU_48) and reduced the relative abundance of *Lactobacillus* genus in the caecum compared to the control and/or 10% IS diets (*p* < 0.05). In Exp. 2, 5% IS supplementation led to compositionally more diverse and different ileal and caecal microbiota compared to the control group (*p* < 0.05; *p* = 0.066 for β-diversity in ileum). Supplementation with the 5% IS increased the relative abundance of *Clostridium celatum*/*disporicum*/*saudiense* (OTU_3) in the ileum and caecum and *Bifidobacterium pseudolongum* (OTU_17) in the caecum and reduced the relative abundance of *Streptococcus gallolyticus*/*alactolyticus* (OTU_2) in the caecum compared to the control diet (*p* < 0.05). Similar effects on *C. celatum*/*disporicum*/*saudiense* and *S. gallolyticus*/*alactolyticus* were observed with the 10% IS diet in the caecum (*p* < 0.05). IS has the potential to beneficially alter the composition of the gastrointestinal microbiota in pigs.

## 1. Introduction

Pig meat is a major animal protein source worldwide, accounting for a third of the overall meat production [1]. However, the intensive pig production systems implemented to achieve high meat production involve efficient feed utilisation, rapid growth rates, and confined rearing in high stocking densities of fattening pigs. These factors exert intense pressure on different components of the gastrointestinal tract, namely the mucosa, the immune system, and the intestinal microbiota, with potential negative impacts on their function and health [2]. The microbiota has a pivotal role in host growth and health via its contributions to nutrient availability, immunomodulation, and inhibition of pathogen colonisation [3,4,5]. However, stressors such as abrupt weaning at an early age, antimicrobial use, and management practices (e.g., diet changes, overcrowding, co-mingling, and poor hygiene) may lead to disturbances in the composition and function of this microbial community, called dysbiosis, predisposing pigs to intestinal infection and reduced productivity [2,6,7,8].

A common practice in the pig industry to control, alleviate, or prevent gastrointestinal dysfunction and disease coupled with modulatory effects on the composition of the microbiota is the use of antimicrobials as treatment, pro- or meta-phylaxis, or even as growth promoters [9,10,11]. However, the emergence of antimicrobial resistance led to the adaptation of stringent measures in antimicrobial use in the European Union, namely the 2006 ban on antimicrobial growth promoters (Regulation—EC: No. 1831/2003) and the restrictions in antimicrobial use for treatment in livestock (Regulations—EU: No. 2019/6 and No. 2019/4), with similar trends also being observed worldwide [12,13,14,15]. An alternative, well-established strategy for manipulating the composition of the gastrointestinal microbiota is via dietary intervention. Agro-industrial waste by-products of plant and animal origin are promising dietary ingredients that are capable of providing energy, protein, and other nutrients, as well as beneficial bioactive components such as polyphenols, nondigestible polysaccharides, and bioactive peptides for livestock. These feed ingredients can be fed to livestock directly or after modification processes (e.g., ensilage or solid-state fermentation) [16,17,18,19]. Furthermore, the utilisation of these locally produced animal feeds, along with the associated reduced costs and environmental impact, contributes to the sustainability of pig production [16,20]. In previous studies, inclusion of various agro-industrial waste by-products in the diet of pigs from different production stages led to changes in the composition of the gastrointestinal and faecal microbiota and the numbers of selected bacterial populations, indicating their ability to influence this microbial community [21,22,23,24,25,26].

In Greece, the production of olive oil, cheese, and wine are important sectors of the agro-industry. However, these activities are associated with the production of agro-industrial waste by-products in large quantities. In an attempt to minimise their environmental burden, an optimised ensilage methodology was developed recently to produce an innovative silage (IS) containing, namely, olive mill wastewater solids, grape pomace solids, and deproteinised feta cheese waste solids, for utilisation in animal nutrition [27]. Subsequent inclusion of the IS in the diets of broiler chickens was associated with improved growth performance, beneficial changes in the gastrointestinal microbiota, predominantly increased *Bifidobacterium* spp., decreased *Enterobacteriaceae* family counts, and improved meat hygiene and quality [28]. The objective of the current study was to provide an initial insight on the effects of two IS inclusion levels in the diets of weaned pigs (Exp. 1) and finishing pigs (Exp. 2) on the composition of the ileal and caecal microbiota using high-throughput amplicon (16S rRNA gene) sequencing.

## 2. Materials and Methods

All procedures described in the two pig trials were approved by the Research Committee of the University of Ioannina, Greece (research project registration: 61291) and were carried out in accordance with Greek (Presidential Degree 56/2013) and EU (Directive 2010/63/EU) legislation regarding the use of animals for scientific purposes. All animals were monitored by a veterinary surgeon throughout the experimental period.

### 2.1. Experimental Design, Diets, and Animal Management

Two consecutive dietary intervention experiments with a randomised complete block design were conducted in weaned pigs (Exp. 1) and finishing pigs (Exp. 2), respectively, at commercial pig farm settings. IS was produced as previously described [27] using the optimal mixture of olive mill waste (60%), grape pomace (20%), and deproteinised feta cheese waste (20%) solids with coarsely ground maize grains after a 30 day fermentation using *Lentilactobacillus* (former *Lactobacillus*) *buchneri* as a starter culture.

In the first experiment (Exp. 1), forty-five crossbred (1/4 Large White × 1/4 Landrace × 1/2 Duroc) weaned pigs (34 days old) were selected (average body of 8.32 ± 1.13 kg) from a commercial pig farm with a 28 day weaning practice and randomly assigned to the three different experimental groups fed with 0% (Control), 5%, or 10% IS, respectively (15 piglets/experimental diet). All pigs were individually ear-tagged. The ingredient composition of the experimental diets and their chemical composition were determined according to AOAC [29] and are presented in Table 1. The experimental diets were isonitrogenous and isocaloric and were formulated to meet the nutrient requirements of the weaned pigs according to the National Research Council guidelines [30]. All pigs in each treatment were housed together in a single fully slated pen and had ad libidum access to feed and water throughout the 40 day experimental period. Commercial management practises were employed regarding artificial lighting and controlling humidity and ambient temperature. The body weight of the pigs was recorded at the start (day 0) and end (day 40) of the trial.

In the second experiment (Exp. 2), eighteen finishing pigs retained from the first experiment were used. In the period between the two experiments (8 weeks), all these pigs were fed the same commercial control diet. On day 97, the eighteen pigs were weighed (average weight 59.47 ± 3.51 kg) and re-grouped according to the initial experimental groups and housed in pens (6 pigs/experimental diet). The ingredient composition of the experimental diets and their chemical composition were determined according to AOAC [29] and are presented in Table 2. The experimental diets were isonitrogenous and isocaloric and were formulated to meet the nutrient requirements of the finishing pigs according to the National Research Council guidelines [30]. All pigs in each treatment were housed together in a single, fully slated pen and had ad libidum access to feed and water throughout the experimental period. Animal management practices were as described in the first experiment. The body weight of the pigs was recorded at the start (day 97) and end (day 157) of the trial.

### 2.2. Digesta Sampling from the Gastrointestinal Tract

At the end of the two pig trials, 15 pigs from Exp. 1 (5 pigs/experimental diet) and 18 pigs from Exp. 2 (6 pigs/experimental diet) were slaughtered under commercial conditions (pre-slaughter electrical stunning; bleeding; scalding/dehairing for Exp. 1 and skinning for Exp. 2; evisceration). For both trials, the gastrointestinal tract of all animals was removed for sampling of ileal and caecal digesta and mucosa. All samples were collected aseptically in sterile containers (Sarstedt AG & Co. KG, Nümbrecht, Germany), homogenised, snap frozen in liquid nitrogen, and stored at −40 °C until further processing.

### 2.3. Assessment of Ileal and Caecal Bacterial Diversity Using High-Throughput Sequencing

#### 2.3.1. DNA Extraction

Microbial genomic DNA was extracted from the ileal and caecal samples using the DNeasy PowerSoil Pro Kit (Qiagen, Hilden, Germany) according to the manufacturer’s instructions. DNA quantity was determined using a Qubit 4 fluorometer (Invitrogen, Thermo Fisher Scientific, Waltham, MA, USA).

#### 2.3.2. 16S rRNA High-Throughput Sequencing

Aliquots of the extracted DNA were used for PCR amplification of the V3–V4 region of the 16S rRNA using the primers 341F and 806R [31]. After the preparation and purification of the amplicon libraries using the Nextera XT index kit (Illumina Inc., San Diego, CA, USA) and the AMPure XP system (Beckmann Coulter, Krefeld, Germany) according to the manufacturer’s instructions and the addition of the PhiX control library (Illumina Inc., San Diego, CA, USA) at a 10% *v/v* ratio, paired-end (300 bp) sequencing was carried out on the MiSeq Sequencing System (Illumina Inc., San Diego, CA, USA) using the MiSeq Reagent Kit v2 (300 cycles) (Illumina Inc., San Diego, CA, USA).

#### 2.3.3. Bioinformatics Analysis

The obtained sequencing data of the 16S rRNA amplicons were further processed using the UPARSE-based “Integrated Microbial Next-generation sequencing” platform (IMNGS, www.imngs.org, accessed on 9 December 2022) [32,33]. Raw sequence reads were demultiplexed (demultiplexer v3.pl), trimmed by ten nucleotides, filtered (sequences with <200 and >600 nucleotide lengths excluded), and merged with paired reads. Clustering of operational taxonomic units (OTUs) was set at a 97% similarity level, while rare OTUs (<0.25% relative abundance) were removed [34]. Alpha and beta diversity and bacterial composition after taxonomic binning were determined using the R script-based Rhea pipeline [35]. Differences in the relative abundances of taxa across dietary treatments were evaluated using the pairwise Wilcoxon Rank Sum test. Similarities to the closest known species for OTUs with significant differences in abundance between dietary treatments were recorded using the 16S-based ID service of EZBioCloud (www.ezbiocloud.net, accessed on 2 February 2023) with the identification similarity percentage provided [36]. Probability values of <0.05 and <0.10 denote statistical significance and tendency, respectively. The individual pig was considered the experimental unit for the statistical analysis. The data were visualised using Illustrator CS6 Version 16.0.0 (Adobe Inc., San José, CA, USA).

### 2.4. Statistical Analysis

Final weight and weight gain data were analysed using the one-way analysis of variance (one-way ANOVA) of SPSS (version 20.0). In both experiments, each individual pig was considered the experimental unit. Levene’s test was used to test data homogeneity, and Tukey’s test was used for post hoc comparisons between the dietary treatments. Statistical significance was set at probability values of 5% (*p* < 0.05).

## 3. Results

### 3.1. Pig Performance

The effects of the different IS inclusion levels in the diets of weaned and finishing pigs on growth performance are presented in Table 3. In both Exp. 1 and Exp. 2, dietary supplementation with 5% and 10% IS had no effect on final body weight or weight gain (*p* > 0.05).

### 3.2. Influence of IS Supplementation on the Composition of the Gastrointestinal Microbiota

To evaluate the effect of the different IS inclusion levels in the diets on the composition of the ileal and caecal microbiota in weaned pigs (Exp. 1) and finishing pigs (Exp. 2), high-throughput sequencing of the 16S rRNA gene (V3–V4 region) was performed. A total of 3,120,963 raw paired-end reads with an average of 49,539 ± 13,467 reads per sample were sequenced from all intestinal samples of both pig trials. Following the post-sequence processing of the reads, the number of high-quality sequences obtained was 2,059,530, with an average of 32,691 ± 9006 reads per sample, and a total of 289 OTUs were observed.

#### 3.2.1. Exp. 1 (Weaned Pigs)

Dietary supplementation with 5% and 10% IS had no effect on the alpha diversity indices (*p* > 0.05) and beta diversity (PERMANOVA *p* > 0.05) in both the ileal (Figure 1A) and caecal (Figure 1B) microbiota of weaned pigs.

In the ileum, the dominant phylum was Firmicutes (93.47%), followed by Actinobacteriota (5.35%), and Pseudomonadota (0.63%), with the remaining phyla (Bacteroidota, Chloroflexota, Verrucomicrobiota, and Spirochaetota) having a relative abundance lower than 0.3%. The most abundant families and genera in the ileal microbiota of the weaned pigs are presented in Figure 2. Interestingly, the *Clostridiaceae* family and *Clostridium* genus were numerically more abundant in pigs fed with the 5% IS diet, whereas the *Lactobacillaceae* family and *Lactobacillus* genus, coupled with the *Limosilactobacillus* genus for the 10% IS diet, were numerically more abundant in the control and the 10% IS-supplemented pigs (*p* > 0.10). However, none of the experimental diets led to statistically significant effects on the composition of the ileal microbiota in weaned pigs (*p* > 0.05).

In the caecum, Firmicutes (68.40%) and Bacteroidota (30.12%) were the predominant phyla, while Actinobacteriota (1.02%), Verrucomicrobiota (0.20%), Pseudomonadota (0.08%), Spirochaetota (0.05%), and Cyanobacteria (0.02%) were minor bacterial populations. The most abundant families and genera in the caecal microbiota of weaned pigs are presented in Figure 3, with statistically significant differences in the relative abundance shown in Figure 4. At the family level, the relative abundance of *Prevotellaceae* increased in pigs fed with the 5% IS diet compared to the control diet (*p* = 0.032), with this observation also occurring as a tendency compared to the 10% IS diet (*p* = 0.056) (Figure 3). At the genus level, the relative abundance of *Coprococcus* increased in pigs fed with the 5% IS diet compared to the control (*p* = 0.036) and 10% IS (*p* = 0.032) diets (Figure 4i), while the relative abundance of *Lactobacillus* decreased in pigs fed with the 5% IS diet compared to the 10% IS diet (*p* = 0.032), with this observation also occurring as a tendency compared to the control diet (*p* = 0.064) (Figure 4ii). At the species level, the relative abundance of *Alloprevotella rava* (OTU_48, 90.43% identification similarity) was increased in pigs fed with the 5% IS diet compared to the 10% IS diet (*p* = 0.032), with this observation occurring as a tendency compared to the control diet (*p* = 0.064) (Figure 4iii).

#### 3.2.2. Exp. 2 (Finishing Pigs)

Dietary supplementation of finishing pigs with the 5% IS led to a compositionally different caecal bacterial community compared to the control diet (PERMANOVA *p* = 0.039, Figure 5B), with this observation also occurring as a tendency for the ileal microbiota (PERMANOVA *p* = 0.066, Figure 5A).

In the ileum, Richness, Shannon Effective, and Simpson Effective indices were increased in pigs fed with the 5% IS diet compared to the control (*p* = 0.041, *p* = 0.009, and *p* = 0.041, respectively) and 10% IS (*p* = 0.017, *p* = 0.052, and *p* = 0.052, respectively), diets indicating a more diverse bacterial community in the ileum of these pigs. The Shannon Effective Index is presented in Figure 6i as a representative of the alpha diversity indices. The dominant phylum was Firmicutes (98.59%), with Actinobateriota (1.38%), Pseudomonadota (0.02%), Bacteroidota (0.01%), and Verrucomicrobiota (<0.001%) being present in low abundance. The most abundant families and genera in the ileal microbiota of the finishing pigs are presented in Figure 7, with statistically significant differences in the relative abundance shown in Figure 6. The relative abundance of the *Clostridiaceae* family (Figure 7), *Clostridium* genus (Figure 6ii), and *Clostridium celatum*/*disporicum*/*saudiense* (OTU_3, 98.33% identification similarity, Figure 6iii) was increased in pigs fed with the 5% IS diet compared to the control diet (*p* = 0.026 for all three comparisons), with this observation also occurring as a tendency compared to the 10% IS diet (*p* = 0.082 for all three comparisons).

In the caecum, the Simpson Effective Index was increased in pigs fed with the 5% and 10% IS diets compared to the control diet (*p* = 0.009 and *p* = 0.017, respectively), indicating that the experimental diets led to a more compositionally diverse caecal microbiota in these pigs (Figure 8i). The dominant phylum was Firmicutes (95.23%), followed by Bacteroidota (2.88%), and Actinobateriota (1.51%), with the remaining phyla (Spirochaetota, Pseudomonadota, Cyanobacteriota, and Verrucomicrobiota) having a relative abundance of less than 0.4%. The most abundant families and genera in the caecal microbiota of the finishing pigs are presented in Figure 9, with statistically significant differences in the relative abundance shown in Figure 8. The relative abundance of *Streptococcaceae* family and *Streptococcus* genus decreased in pigs fed with the 5% (*p* = 0.004 for both comparisons) and 10% (*p* = 0.017 for both comparisons) IS diets compared to the control diet (Figure 8ii and Figure 9), while the relative abundance of *Clostridiaceae* family and *Clostridium* genus increased in pigs fed with the 5% (*p* = 0.004 for both comparisons) and 10% (tendency, *p* = 0.052 for both comparisons) IS diets compared to the control diet (Figure 8iii and Figure 9). These observations were also evident at the species level, with the relative abundance of *Streptococcus gallolyticus*/*alactolyticus* (OTU_2, 100% identification similarity, Figure 8v) decreasing and that of *Clostridium celatum*/*disporicum*/*saudiense* (OTU_3, 98.33% identification similarity, Figure 8vi) increasing for 5% (*p* = 0.004 for both comparisons) and 10% (*p* = 0.017 and *p* = 0.004, respectively) IS diets compared to the control diet. Furthermore, the relative abundance of *Bifidobacterium pseudolongum* (OTU_17, 99.77% identification similarity, Figure 8iv) increased in pigs fed with the 5% IS diet compared to the control (*p* = 0.030) and 10% IS (*p* = 0.009) diets.

## 4. Discussion

Agro-industrial waste by-products are promising feed ingredients associated with more sustainable and cost-effective livestock production and a reduced environmental burden [16]. However, prior research on their dietary and/or nutraceutical potential by investigating their impact on animal performance and health is essential. Previously, an innovative silage containing olive mill wastewater, grape pomace, and deproteinised feta cheese waste solids, major waste by-products of the Greek agro-industry, was developed and utilised in broiler nutrition with beneficial effects on performance and parameters of health and meat quality [27,28]. The current small-scale study aimed to provide insight on the potential of this IS as a diet ingredient for pigs by investigating its effects on the performance and composition of the gastrointestinal microbiota of weaned (Exp. 1) and finishing (Exp. 2) pigs.

In the present study, the IS addition at two inclusion levels (5% and 10%) had no effect on body weight or weight gain in either of the two experiments with the weaned and finishing pigs. Previous studies evaluating the use of olive mill, winery, or cheese-making waste by-products as feed ingredients in the diet of weaned, growing, and finishing pigs mostly reported no significant effects on performance parameters [23,25,37,38,39,40], except for one report [21], in which grape pomace-containing silage supplementation increased several of these parameters in weaned pigs. In the initial investigation of the same IS as a feed additive in broiler diets, the body weight and weight gain of the chicken significantly increased with the 10% inclusion level [28], but this effect was not replicated in the present study with the pigs. This difference in efficacy highlights the differences in digestive physiology between poultry and swine.

Considering the important role of the gastrointestinal microbiota on animal health and productivity, this study focused on whether IS could influence the composition of the microbial community in weaned and finishing pigs. Although several studies evaluating the use of olive mill, winery, or cheese-making waste by-products as feed ingredients in pig nutrition reported compositional changes in the gastrointestinal microbiota, the majority of them utilised culture-dependent PCR targeting specific bacterial populations in faeces [21,24,25,37,41], with only two of them performing a more in-depth investigation of the caecal and faecal microbial community using 16S rRNA high-throughput sequencing [23,39]. In the current work, the latter methodology was employed in an attempt to provide a better insight into the alterations induced by IS in the ileal and caecal microbiota.

In Exp. 1, IS inclusion in the diet of weaned pigs had a limited effect, mostly evident on the caecal microbiota. In particular, no significant differences were observed in the ileal microbiota; however, the 5% IS-fed pigs were characterised by a numerically higher relative abundance of the *Clostridiaceae* family and *Clostridium* genus, while the *Lactobacillaceae* family and *Lactobacillus* and *Limosilactobacillus* genera were numerically more abundant in the control and 10% IS groups. This observation might indicate that the 5% inclusion level promoted a faster maturation of the ileal microbiota post-weaning [42]. This notion is further supported by the compositional changes that occurred in the caecal microbiota of the 5% IS-fed pigs. Namely, the relative abundances of the *Prevotellaceae* family and *Coprococcus* genus increased while the relative abundance of the *Lactobacillus* genus decreased, all of which represent well-established shifts in the maturing microbiota [43,44,45,46,47]. This finding could be attributed to the flavonoids contained in the IS, as similar changes have been observed in the faecal microbiota of weaned pigs supplemented with quercetin, a flavonoid compound that is also present in winery and olive mill waste by-products [48,49]. The *Prevotellaceae* family and *Coprococcus* genus are involved in the fermentation of nondigestible carbohydrates and the production of short-chain fatty acids, as well as other important functions such as immunomodulation and colonisation resistance to pathogens, all of which contribute to the improved pig performance associated with these bacterial populations [43,47,50,51,52]. Furthermore, the inclusion of the 5% IS in the diet of weaned pigs led to an increase in the relative abundance of OTU_48, identified as a member of the *Alloprevotella* genus (90.43% similarity to *A. rava*), a typical coloniser of the caecal mucosa associated with nondigestible fibre degradation and propionate production [53,54,55]. Based on the above, the IS seems to have accelerated the maturation of the gastrointestinal microbiota by stimulating bacterial populations involved in the utilisation of complex carbohydrates and the production of short-chain fatty acids, rendering it a promising feed ingredient post-weaning that should be further explored in future studies.

Dietary supplementation with IS led to an alteration in the composition of the ileal and caecal microbiota of finishing pigs, with the 5% inclusion level exerting the strongest effect. In terms of α- and β-diversity, the ileal and caecal microbiota were compositionally more diverse and different in the 5% IS-fed pigs compared to the control group. Inclusion of 10% IS was also associated with increased diversity in the caecal microbiota. The main compositional changes induced by IS supplementation included an increase in the relative abundances of *Clostrdiaceae* family, *Clostridium* genus, and *C. celatum*/*disporicum*/*saudiense* (OTU_3, 98.33% identification similarity) in the ileum and caecum and a reduction in the relative abundances of *Streptococcaceae* family, *Streptococcus* genus, and *S. gallolyticus*/*alactolyticus* (OTU_2, 100% identification similarity) in the caecum. Similar observations were previously reported in weaned and growing pigs following dietary inclusion of quercetin and polyphenol-rich grape processing by-products, indicating that the observed changes are likely associated with the phenolic compounds of the olive mill and winery waste by-products of the IS [25,48,49]. *C. celatum* has previously been considered to contribute to carbohydrate metabolism, mucin degradation, and SCFA production as well as to cross-feeding by producing simpler carbohydrates and peptides in vitro and in pigs, whereas the increase of *C. disporicum*/*saudiense* was negatively correlated to growth and butyrate production in pigs [48,56,57], indicating that further research concerning the role of these clostridial species in pigs is required. *S. gallolyticus*/*alactolyticus* are members of the *S. bovis*/*S. equinus* complex, common colonisers of the gastrointestinal tract involved in protein and carbohydrate degradation and opportunistic pig pathogens with an emerging role in bacterial endocarditis [58,59,60], that have been considered signatures of the gastrointestinal dysbiosis induced by *Salmonella enterica* subsp. *enterica* serovar Typhimurium and porcine epidemic diarrhoea virus infections in pigs [61,62]. An additional finding was the increase in the relative abundance of *Bifidobacterium pseudolongum* (OTU_17, 99.77% identification similarity) in the caecum with IS supplementation, with similar observations reported in previous studies evaluating the use of olive mill, winery, or cheese-making waste by-products in pig diets associated with the nondigestible polysaccharides contained in these ingredients [21,23,24]. *B. pseudolongum* has been shown to possess a wide-ranging repertoire of carbohydrate-degrading enzymes as well as to exert immunomodulatory effects via direct interaction with the host or compositional manipulation of the gastrointestinal microbiota. Thus, IS inclusion in the diet of the finishing pigs was associated with a reduction of opportunistic pathogens and stimulation of commensals with potentially beneficial functions in pig nutrition and health that merit further research.

## 5. Conclusions

In conclusion, IS addition in the diets led to alteration in the ileal and caecal microbiota of weaned and finishing pigs, with the 5% inclusion level considered to be most effective. In the case of weaned pigs, IS supplementation was associated with an increase in the *Prevotellaceae* family, *Coprococcus*, and *Alloprevotella* genera and a reduction in the *Lactobacillus* genus in the caecum, indicative of a more mature microbiota capable of complex polysaccharide utilisation. In finishing pigs, IS supplementation increased *C. celatum*/*disporicum*/*saudiense* and *B. pseudolongum* and reduced *S. gallolyticus*/*alactolyticus*, suggesting an enhanced carbohydrate metabolism and improved health in the gastrointestinal tract. The findings of this small-scale study show that innovative silage is a promising feed ingredient for use in pig nutrition that should be further investigated regarding its effects on performance and gut functionality and health.

## Figures and Tables

**Figure 1 microorganisms-11-01723-f001:**
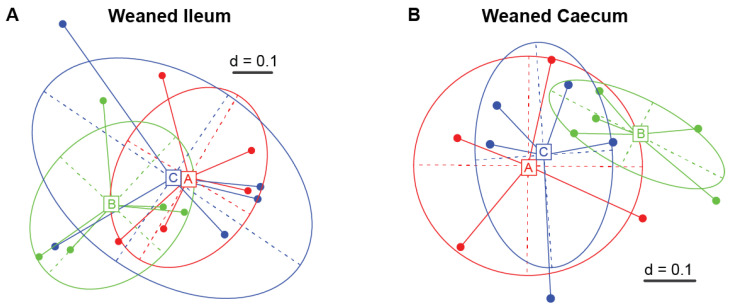
Multidimensional scaling (MSD) plots based on the generalised UniFrac dissimilarity matrix of ileal (**A**) and caecal (**B**) microbial profiles of each dietary treatment group of weaned pigs. There was no statistical significance (*p* > 0.05) for all pairwise comparisons of the dietary treatment groups based on the PERMANOVA test. A total of five replicates were used per experimental diet (replicate = pig). A, control; B, 5% IS; and C, 10% IS.

**Figure 2 microorganisms-11-01723-f002:**
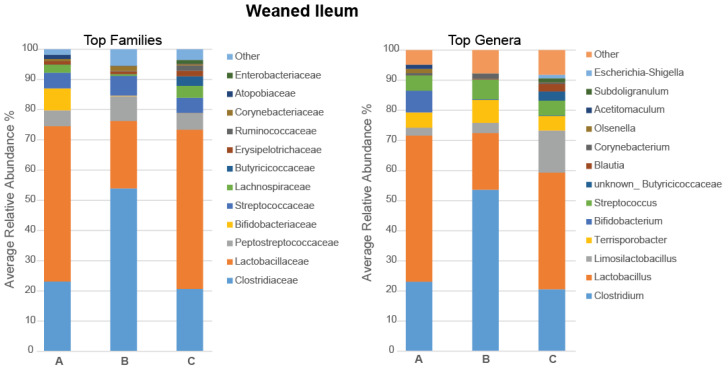
Stacked bar plots of the average relative abundance of the top bacterial families and genera in the ileal microbiota of the weaned pigs fed with the control (A), 5% IS (B), and 10% IS (C) diets. A total of five replicates were used per experimental diet (replicate = pig).

**Figure 3 microorganisms-11-01723-f003:**
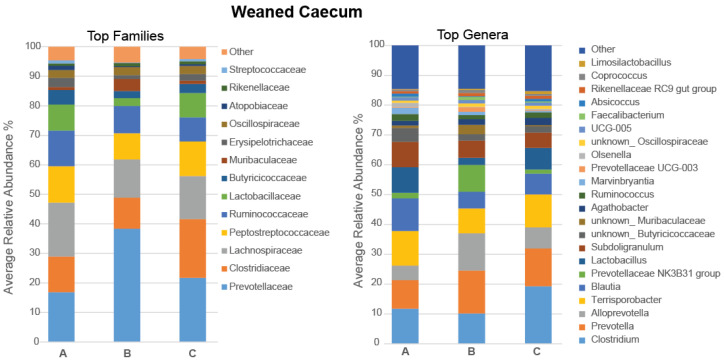
Stacked bar plots of the average relative abundance of the top bacterial families and genera in the caecal microbiota of the weaned pigs fed with the control (A), 5% IS (B), and 10% IS (C) diets. A total of five replicates were used per experimental diet (replicate = pig).

**Figure 4 microorganisms-11-01723-f004:**
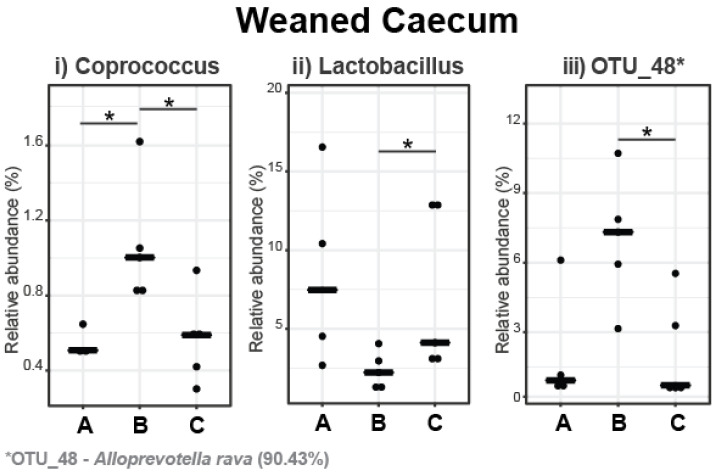
Differential relative abundance of bacterial genera (**i**,**ii**) and OTUs (**iii**) in the caecal microbiota in response to IS supplementation in weaned pigs. The bold line represents the median, while the * symbol indicates statistically significant differences in the relative abundance between dietary treatment groups (Wilcoxon Rank Sum statistical test), with the number of stars indicating the level of significance (* < 0.05). A total of five replicates were used per experimental diet (replicate = pig). A, control; B, 5% IS; and C, 10% IS.

**Figure 5 microorganisms-11-01723-f005:**
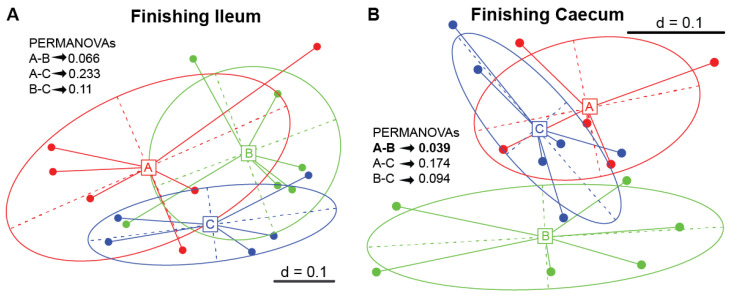
Multidimensional scaling (MSD) plots based on the generalised UniFrac dissimilarity matrix of ileal (**A**) and caecal (**B**) microbial profiles of each dietary treatment group of finishing pigs. PERMANOVA pairwise comparisons of the dietary treatment groups with a *p*-value < 0.05 denote statistically significant differences. A total of six replicates were used per experimental diet (replicate = pig). A, control; B, 5% IS; and C, 10% IS.

**Figure 6 microorganisms-11-01723-f006:**
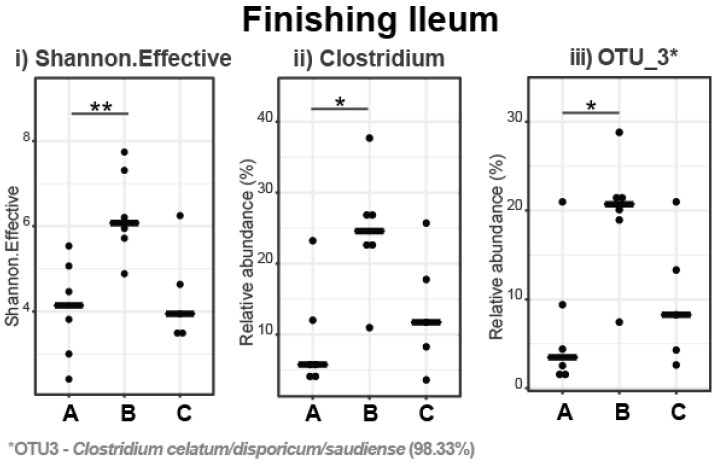
Differences in alpha diversity indices (**i**) and relative abundance of bacterial genera (**ii**) and OTUs (**iii**) in the ileal microbiota in response to IS supplementation in finishing pigs. The bold line represents the median, while the * symbol indicates statistically significant differences in the Shannon Effective Index and the relative bacterial abundance between dietary treatment groups (Wilcoxon Rank Sum statistical test), with the number of stars indicating the level of significance (* < 0.05, ** < 0.01). A total of six replicates were used per experimental diet (replicate = pig). A, control; B, 5% IS; and C, 10% IS.

**Figure 7 microorganisms-11-01723-f007:**
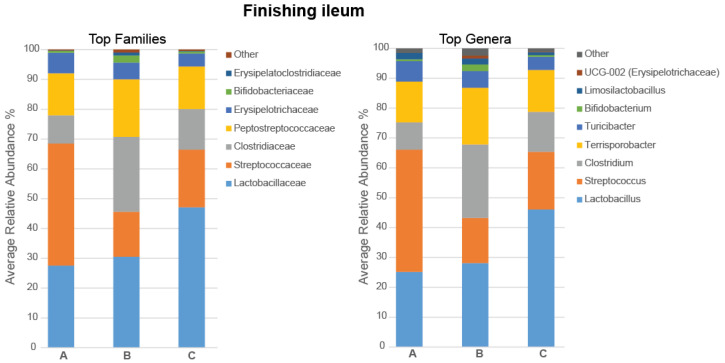
Stacked bar plots of the average relative abundance of the top bacterial families and genera in the ileal microbiota of the finishing pigs fed with the control (A), 5% IS (B), and 10% IS (C) diets. A total of six replicates were used per experimental diet (replicate = pig).

**Figure 8 microorganisms-11-01723-f008:**
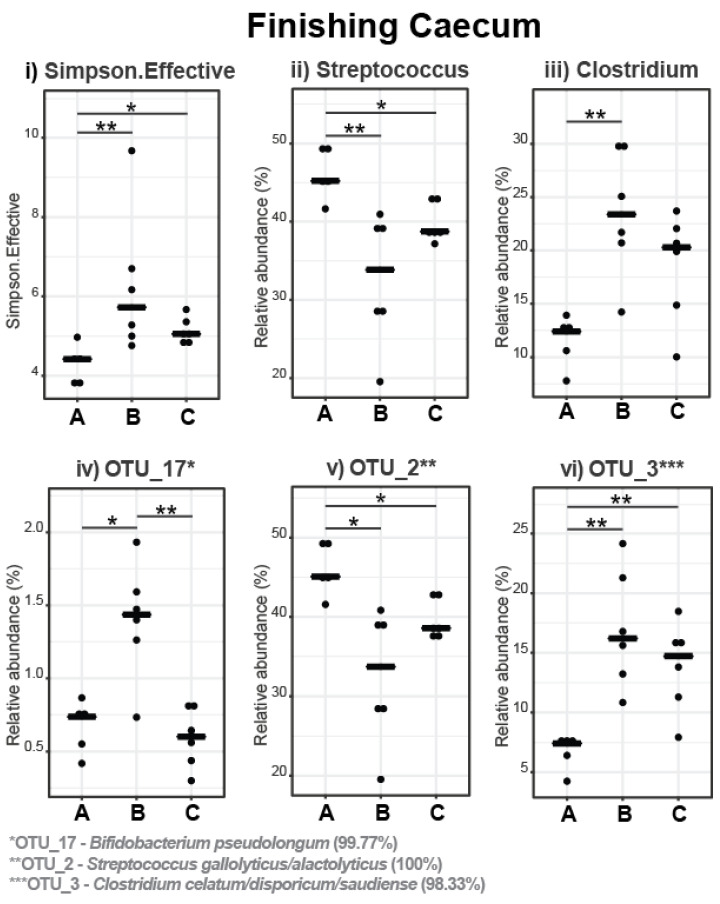
Differences in alpha diversity indices (**i**) and relative abundance of bacterial genera (**ii**,**iii**) and OTUs (**iv**–**vi**) in the caecal microbiota in response to IS supplementation in finishing pigs. The bold line represents the median, while the * symbol indicates statistically significant differences in the Simpson Effective Index and the relative bacterial abundance between dietary treatment groups (Wilcoxon Rank Sum statistical test), with the number of stars indicating the level of significance (* < 0.05, ** < 0.01). A total of six replicates were used per experimental diet (replicate = pig). A, control; B, 5% IS; and C, 10% IS.

**Figure 9 microorganisms-11-01723-f009:**
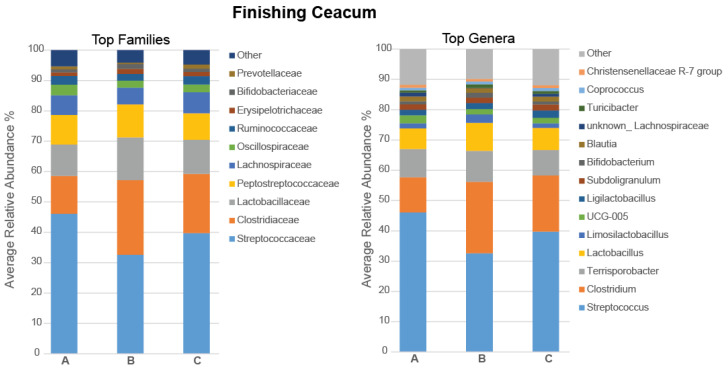
Stacked bar plots of the average relative abundance of the top bacterial families and genera in the caecal microbiota of the finishing pigs fed with the control (A), 5% IS (B), and 10% IS (C) diets. A total of six replicates were used per experimental diet (replicate = pig).

**Table 1 microorganisms-11-01723-t001:** Composition and chemical analysis of the control and experimental diets of Exp. 1.

Ingredient (%)	Weaned Pig Diets		
	Control	5% IS	10% IS
Maize	43.60	36.97	30.34
Innovative Silage (IS)	0.00	5.00	10.00
Soybean meal (47% CP)	15.80	16.37	16.94
Barley	20.00	20.00	20.00
Fishmeal (62% CP)	3.00	3.00	3.00
Wheat middlings	3.00	3.00	3.00
Soybean oil	2.00	3.06	4.12
Minerals and vitamins premix *	6.00	6.00	6.00
Whey permeate (4.5% CP)	6.00	6.00	6.00
Zinc oxide	0.30	0.30	0.30
Benzoic acid	0.30	0.30	0.30
Chemical analysis (as fed basis)			
Digestible energy, MJ/kg	13.87	13.86	13.86
Crude protein, %	17.64	17.64	17.64
Dry matter, %	89.03	87.68	86.32
Ash, %	5.45	5.45	5.46
Crude fat, %	4.50	5.43	6.37
Crude fibre, %	2.90	2.86	2.82
Acid detergent fibre, %	3.30	3.26	3.22
Neutral detergent fibre, %	9.86	9.61	9.36
Ca, %	0.58	0.58	0.58
Total P, %	0.49	0.48	0.48
Lysine, %	1.18	1.19	1.19
Methionine + Cystine, %	0.74	0.74	0.73

* Provided per kg of complete diet: 15,000 IU retinyl acetate, 50 mcg 25-hydroxycholecalciferol, 9.96 mg alpha-tocopherol acetate, 10.02 mg menadione nicotinamide bisulphite, 3 mg thiamine mononitrate, 10.02 mg riboflavin, 6 mg pantothenic acid, 6 mg pyridoxine hydrochloride, 40.02 mcg cyanocobalamin, 100 mg ascorbic acid, 35 mg niacin, 300 mcg biotin, 1.5 mg folic acid, 375 mg choline chloride, 200 mg ferrous sulphate monohydrate, 90 mg copper sulphate pentahydrate, 60 mg manganese sulphate monohydrate, 100 mg zinc sulphate monohydrate, 2 mg calcium iodate, 300 mg sodium selenide, 150 mg L-selenomethionine—selenium, 1500 FYT 6-phytase, 80 U β-1,4-endoglucanase, 70 U β-1,3 (4)-endoglucanase, 270 U β-1,4-endoxylanase, 5000 mg benzoic acid, 40.8 mg butylated hydroxy-toluene, and 3.5 mg propyl gallate.

**Table 2 microorganisms-11-01723-t002:** Composition and chemical analysis of the control and experimental diets of Exp. 2.

Ingredient (g/kg)	Finishing Pig Diets		
	Control	5% IS	10% IS
Wheat	45.00	37.45	30.05
Innovative silage (IS)	0.00	5.00	10.00
Soybean meal (47% CP)	14.00	15.60	17.10
Barley	27.60	27.60	27.60
Wheat middlings	9.50	9.50	9.50
Soybean oil	1.15	2.10	3.00
Minerals and vitamins premix *	1.00	1.00	1.00
Booster supplement ^†^	0.25	0.25	0.25
Salt	0.50	0.50	0.50
Limestone	1.00	1.00	1.00
Chemical analysis (as fed basis)			
Digestible energy, MJ/kg	13.34	13.34	13.34
Crude protein, %	16.55	16.75	16.95
Dry matter, %	87.68	86.33	84.98
Ash, %	4.56	4.57	4.58
Crude fat, %	2.43	3.38	4.33
Crude fibre, %	3.88	3.86	3.84
Acid detergent fibre, %	4.62	4.59	4.56
Neutral detergent fibre, %	13.65	13.45	13.24
Ca, %	0.57	0.57	0.57
Total P, %	0.39	0.39	0.39
Lysine, %	1.07	1.10	1.13
Methionine + Cystine, %	0.62	0.62	0.62
Threonine, %	0.75	0.76	0.78
Tryptophan, %	0.23	0.23	0.24

* Provided per kg complete diet: 6500 IU retinyl acetate; 1200 IU cholecalciferol; 12.5 mcg 25-hydroxycholecalciferol; 60 mg alpha-tocopherol acetate; 2 mg menadione nicotinamide bisulphite; 2 mg thiamine mononitrate; 7 mg riboflavin; 25 mg pantothenic acid; 3 mg pyridoxine hydrochloride; 25 mcg cyanocobalamin; 25 mg nicotinic acid; 1 mg folic acid; 0.15 mg biotin; 300 mg choline chloride; 108 mg Fe from ferrous sulphate monohydrate; 25 mg Cu from copper sulphate; 48 mg Mn from manganese oxide; 84 mg Zn from zinc oxide; 1.2 mg I from calcium iodate; 0.24 mg Se from sodium selenite; 700 mg methionine; 100 mg L-tryptophan; 2730 L-Lysine mg HCl; 1182.02 mg L-threonine; 1500 FYT 6-fytase; 200 FXU endo-1,4-β-xylanase. ^†^ Provided per kg of complete diet: 871.88 mg L-lysine HCl; 824.74 mg L-threonine; 98.87 mg L-tryptophan; and 44 mg DL-methionine.

**Table 3 microorganisms-11-01723-t003:** Effect of IS supplementation on the performance of weaned and finishing pigs.

Weaned Pigs (Exp. 1)	Control	5% IS	10% IS	SEM	*p*-Value
Initial body weight *, kg	8.30	8.32	8.40	0.177	0.975
Final body weight *, kg	26.47	26.38	27.64	0.494	0.518
Weight gain *, kg	18.16	18.06	19.24	0.391	0.401
**Finishing pigs (Exp. 2)**					
Initial body weight ^†^, kg	57.75	59.48	61.18	0.850	0.272
Final body weight ^†^, kg	122.08	123.60	127.95	1.510	0.281
Weight gain ^†^, kg	64.33	64.11	66.76	1.250	0.659

* *n* = 45 (15 pigs per experimental diet); ^†^ *n* = 18 (6 pigs per experimental diet); IS, innovative silage; and SEM, Standard Error of Mean.

## Data Availability

Primary sequencing data were uploaded to the ENA public repository with the accession number PRJEB61448.

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
