# Peer review of "Use of an Innovative Silage of Agro-Industrial Waste By-Products in Pig Nutrition: A Pilot Study of Its Effects on the Pig Gastrointestinal Microbiota"

_microorganisms, 2023, doi:10.3390/microorganisms11071723_

Round 1

Reviewer 1 Report

This is innovation research about agro-industrial waste by products in pig feeds and feeding, and there is very novelty. The experimental design of this study is reasonable, but the background is less and not clear in introduction parts.  To sum up, full manuscript could be accepted after minor revision.

Minor editing of English language required.

Author Response

Response to Reviewer 1:

Comment: This is innovation research about agro-industrial waste by products in pig feeds and feeding, and there is very novelty. The experimental design of this study is reasonable, but the background is less and not clear in introduction parts.  To sum up, full manuscript could be accepted after minor revision. Comments on the Quality of English Language. Minor editing of English language required.

Response: We would like to thank Reviewer 1 for the effort and the positive response. We revised our manuscript based on all the suggestions of the Reviewers to improve the language and the overall quality.

Reviewer 2 Report

Use of innovative silage of agro-industrial waste by-product in pig nutrition: a pilot study of its effects on the gastrointestinal microbiota. The low number of phenotypical and microbiota data observations raises concerns about the repeatable.   

Do we know the tannin and polyphenol concentrations of IS products?

Line 110: How many pigs were housed in a pen, and what is the size of the pen? The same question is for trial 2.

Table 3 Please include the initial BW in the table.

Figures 1,2and 3, What is the possible cause of differences between 5% IS and 10% IS? IF phenolic compound drives microbiota modulation, why are the results of 10% IS similar to the control?

Line 67 Eu also phase out the pharmaceutical Zinc usage in nursery pigs. In addition, data also showed that pharmaceutical Zinc and organic acids can alter the microbiota. That means the results of this trial are confounded with pharmaceutical Zinc and organic acids.

Line 381: Are age-associated microbiota shared between segments?

Author Response

Response to Reviewer 2:

Comment 1: Use of innovative silage of agro-industrial waste by-product in pig nutrition: a pilot study of its effects on the gastrointestinal microbiota. The low number of phenotypical and microbiota data observations raises concerns about the repeatable.  

Response 1: We would like to thank the Reviewer for the effort to review our manuscript.

Regarding the number of observations, we would like to response that this was a pilot study on the use of this novel feed ingredient. The number of animals in this study was selected based on the available amount of silage that was produced and available to test. However, in both trials all animals were individually marked and therefore were considered as separate experimental units for the purpose of the microbiota analysis study and the statistical analysis of the results. We would additionally like to mention that future studies will be carried out in which much larger amounts of the silage will be produced, to be tested in large-scale trials.

Comment 2: The low number of phenotypical and microbiota data observations raises concerns about the repeatable. 

Response 2: Thank you for your comment, we are aware that this study had a low number of replicates per treatment group, however, this was meant to be an initial investigation of the applicability of agro-industrial waste by-products in pigs’ nutrition.

Comment 3:

Do we know the tannin and polyphenol concentrations of IS products?

Response 3:

All details concerning the design and production of the examined silage and its composition are available in the following published paper: Petrotos, K.; Papaioannou, C.; Kokkas, S.; Gkoutsidis, P.; Skoufos, I.; Tzora, A.; Bonos, E.; Tsinas, A.; Giavasis, I.; Mitsagga, C. Optimization of the Composition of a Novel Bioactive Silage Produced by Mixing of Ground Maize Grains with Olive Mill Waste Waters, Grape Pomace and Feta Cheese Whey. AgriEngineering 2021, 3, 868-893, doi:10.3390/agriengineering3040055. Unfortunately, no further characterization regarding the components of this silage was carried out. However, this will be explored in future studies.

Comment 4:

Line 110: How many pigs were housed in a pen, and what is the size of the pen? The same question is for trial 2.

Response 4:

All pigs of each treatment group were placed in a single pen. We modified the materials and methods section of both trials to clarify this.

---------------------

Comment 5:

Table 3 Please include the initial BW in the table.

Response 5:

We added this for both trials as requested.

---------------------

Comment 6:

Figures 1, 2 and 3, What is the possible cause of differences between 5% IS and 10% IS? IF phenolic compound drives microbiota modulation, why are the results of 10% IS similar to the control?

Response 6:

We would like to thank the reviewer for the comment. As stated in comment 3, no further compositional characterization of the dietary supplement was performed, rendering difficult the identification of the specific reasons leading to the observed differences.

Potential explanations for the lack of effect of the 10% dietary supplement compared to the 5% level, include differences in animal appetite and palatability of the experimental diets, the influence of the different levels of the dietary supplement on ingestion, digestion and intestinal absorption and the presence of anti-nutritional factors or other bioactives or components that nullify the effects of the supplement when it is used at higher concentrations.

In future experiments, we will evaluate lower doses up to 5%, and higher doses up to 15%, as this product provide a cheap and affordable source of nutrients aiming to determine the optimal inclusion level in the diet. Additionally, the composition of this dietary supplement will be further explored to determine the bioactive ingredients of this supplement.

Comment 7:

Line 67 Eu also phase out the pharmaceutical Zinc usage in nursery pigs. In addition, data also showed that pharmaceutical Zinc and organic acids can alter the microbiota. That means the results of this trial are confounded with pharmaceutical Zinc and organic acids.

Response 7:

We thank the Reviewer for this insightful comment. Please consider that due to our mistake the footnotes of tables 1 and 2 did not correctly shown the values for the included feed additives (Mineral and vitamin premix; Booster) “per kg of complete diet” but instead “per kg of the product”. The actual values “per kg of feed” were much lower. In the revised manuscript, we corrected the two footnotes. In addition, we agree that the limits for Zinc have become much stricter in recent years. Still the amounts used in trials (in 2022) were within acceptable limits, according to local legislation. Moreover, this study aimed to examine the new feed ingredient under commercial conditions and therefore the recommended levels for the vitamin & mineral premixes were included in the two diets according to the production stages of pigs.

Comment 8:

Line 381: Are age-associated microbiota shared between segments?

Response 8:

Thank you for this comment. If we have understood correctly, the reviewer is referred to the shift of the gut microbiota during the weaning transition which is characterized by a decrease in the milk-orientated bacterial groups and an increase in bacterial groups utilizing complex carbohydrates. This shift is observed in all segments of the gut without bacterial groups necessarily being shared among them at the species level. In terms of the findings in experiment 1, the observed changes in the gut microbiota of the pigs fed with the 5% dietary supplement indicate that this supplement may have contributed to this naturally occurring maturation process. However, future studies are needed to confirm this assumption.

Reviewer 3 Report

The paper deals with the  use of an innovative silage of agro-industrial waste by-products in pig nutrition. Subject of article is current and important because composition of the feed mixture influences the successful production of pig, which is important from the point of view of economy and efficiency of breeding. The results of the research obtained have been properly described and discussed and provide some new findings.

Author Response

Response to Reviewer 3:

Comment:

The paper deals with the use of an innovative silage of agro-industrial waste by-products in pig nutrition. Subject of article is current and important because composition of the feed mixture influences the successful production of pig, which is important from the point of view of economy and efficiency of breeding. The results of the research obtained have been properly described and discussed and provide some new findings.

Response:

We would like to thank Reviewer 3 for the effort and the positive response. We revised our manuscript based on all the suggestions of the Reviewers to improve the language and the overall quality.